# Model of Nano-Metal Electroplating Process in Trapezoid Profile Groove

**Oksana Yu. Egorova [1],\*, Victor G. Kosushkin [2],\* and Leo V. Kozhitov [2],\***

[1]   Kaluga Branch of Bauman Moscow State Technical University, Bazhenov St., 248001 Kaluga, Russia
[2]   Moscow Institute of Steel and Alloys, Leninsky Prospect 2, 196036, Russia
\*   Correspondence: oksana.egorova95@yandex.ru (O.Y.E.); kosushkin@gmail.com (V.G.K.);
     kozitov@rambler.ru (L.V.K.)

**Abstract:** The principle of the electrodeposition method is to immerse the coated products in a water electrolyte solution, the main components of which are salts or other soluble compounds—metal coatings. The software COMSOL Multiphysics was allowed to perform a simulation of the processes of electrodeposition of the metals copper and silver in the groove of the trapezoidal profile.

**Keywords:** electrodeposition process; group 1 metals; modeling; COMSOL Multiphysics

---

## 1. Introduction

The process of electrodeposition of metals is important in micro- and nano-electronics, as it is used in the production of multilayer printed circuit boards (MPC). MPCs consist of many layers, most of which are complex electrical circuits. The formation of interlayer compounds is an important technological task and, as a rule, is carried out with the help of through metallized holes [1]. The manufacture of transition holes with diameters of less than 300 microns with the help of various metallized pastes is complicated by the difficulties of free penetration of the paste into the hole and the air out of it. The simplicity, availability, and technological capabilities of the electrodeposition process make it possible to use it for local electrochemical deposition, especially with an unchanged decrease in topological dimensions. Electrodeposition is a complex process occurring at the interface of type 1 and 2 conductors and depends on various factors such as temperature, mixing rate, and electrolyte composition, as well as ion solvation processes, adsorption at the phase boundary, the state of the double electric layer, the phenomena of electrode polarization, diffusion, and convection flows near the deposition surface. However, the process of local metallization is affected by a complex surface relief.

In some cases, mathematical modeling methods can replace a full-scale experiment, saving material and time costs. In this article, we consider the possibility of mathematical modeling of electrochemical deposition of copper and silver under local metallization in a hollow trapezoidal profile groove using the basic package of COMSOL Multiphysics.

## 2. The Process of Electrodeposition

The principle of the electrodeposition method is to immerse the coated products in an aqueous electrolyte solution. The main components are salts or other soluble compounds called metal plating. The coated products are in contact with the negative pole of the direct current source, i.e., the cathode. A plate made of a deposited metal or an inert electrically conductive material, such as graphite, is used as the anode. It is in contact with the positive pole of the direct current source and dissolves when the flow of electric current, compensating for the decrease of ions, is discharged on the coated products [2]. The reduction of cations on the cathode leads to the release of a metal layer on the negative surface of the cathode. As a result, a metal film with a fine-crystalline, coarse-crystalline, shiny, or matte surface

structure can be formed on the cathode. The deposition of metal atoms begins at the crystallization centers, the number of which is determined by the amount of overvoltage at the cathode. Defects in the structure of the substrate influence the nucleation process, and then they move along the surface to the fractures, forming a film. Thus, the film develops into islands that grow in all directions until they merge [3].

### 3. Modeling the Process of Electroplating Metals of the 1st Group in a Trapezoid Profile Groove Using the Software COMSOL Multiphysics

The deposition of metals in the trapezoid profile groove is an example of the modeling of the electrodeposition process in the COMSOL Multiphysics program. The standard model Copper Deposit in a Trench was taken as a basis [4].

In this paper, we used a trapezoidal profile groove to show how the electrodeposition process proceeds depending on the change in form and composition.

The geometry of the model is shown in Figure 1. The upper horizontal boundary is the anode, and the trapezoidal cathode is located at the bottom.

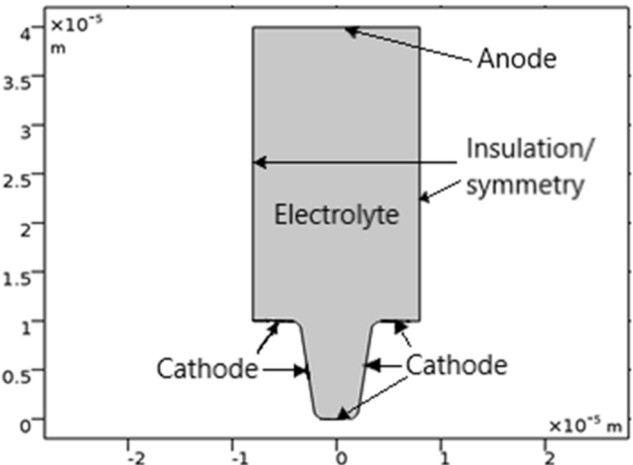

**Figure 1.** The geometry of the model.

The vertical walls on the main electrode are considered isolated:

$$N_i \times n = 0 \tag{1}$$

The flow for each of the ions in the electrolyte is determined by the Nernst–Planck equation:

$$N_i = -D_i \nabla c_i - z_i u_i F c_i \nabla \varphi_i \tag{2}$$

where $N_i$ is the ion flow (mol/m$^2$·s), $c_i$ is the ion concentration in the electrolyte (mol/m$^3$), $z_i$ is the charge of the ion particles, $u_i$ is themobility of charged particles (m$^3$/s·J·mol), $F$ is theFaraday constant (As/mol) is the potential in the electrolyte (V).

This equation allows us to calculate the distribution of copper ion concentrations, isopotential lines, current density lines, and displacements of the cathode and anode surfaces after 12, 14, 16, and 20 s of operation, which are shown in Figure 2.

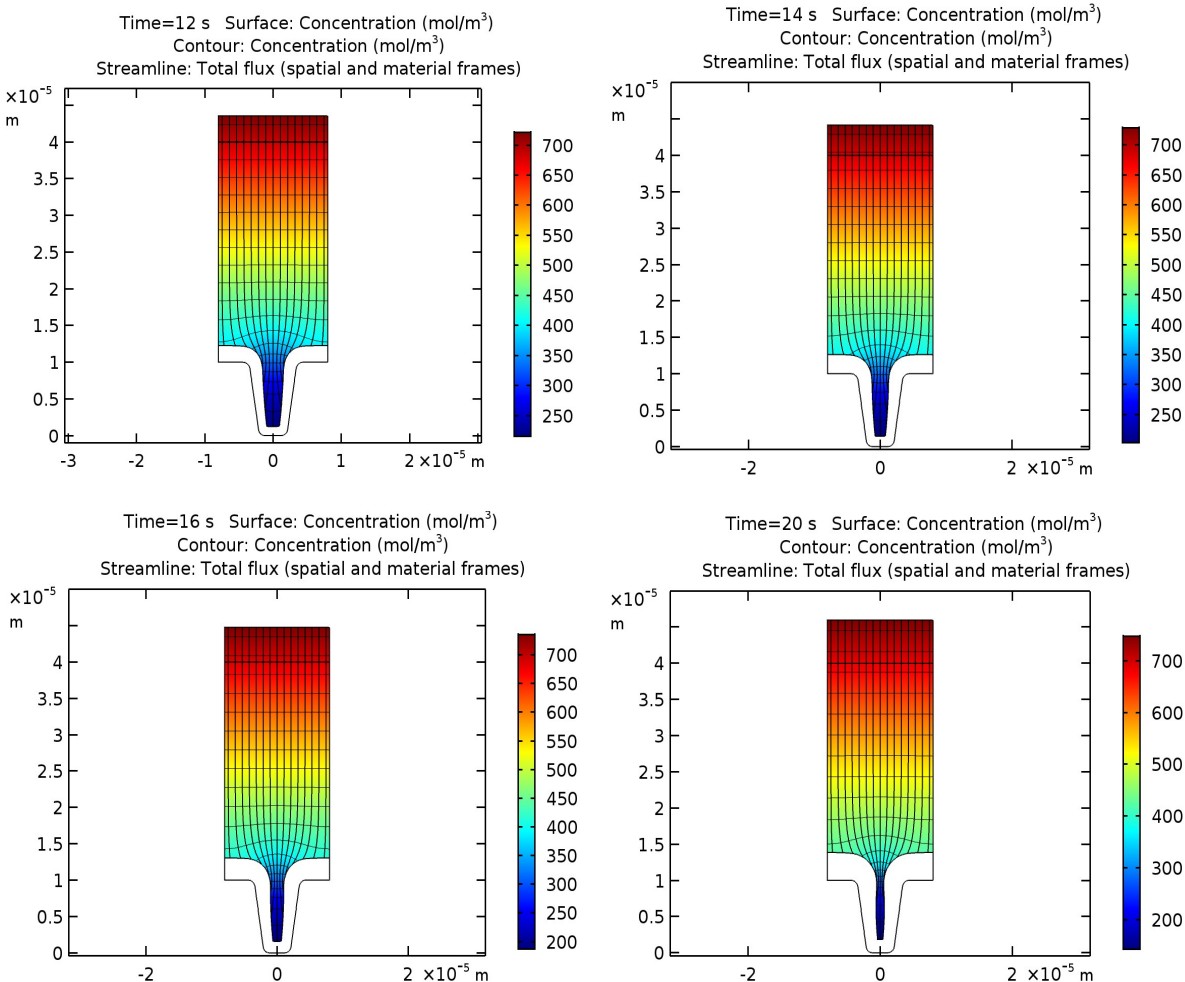

**Figure 2.** Concentrations of copper ions (mol/m³), isopotential lines, current density lines, and electrode displacement in the cell after 12, 14, 16, and 20 s of work.

Lines of layer thickness show the development of non-uniform deposition of a copper layer due to the non-uniform distribution of current density from the bottom to the upper edge of the groove. This effect is directly related to the decrease in the concentration of copper ions in the depth of the cavity.

From this, it can be concluded that with an increase in the duration of the process, the groove hole on the upper edge begins to taper, and its overgrowth is possible due to the non-uniform deposition thickness. Therefore, for copper deposition, the optimal process duration is 14 s.

An analogical modeling was performed for the silver deposition on the surface of the trapezoidal profile.

Figure 3 shows the distribution of the concentration of silver ions, isopotential lines, current density lines, and the displacement of the cathode and anode surface after 12, 14, 16, and 20 s of operation.

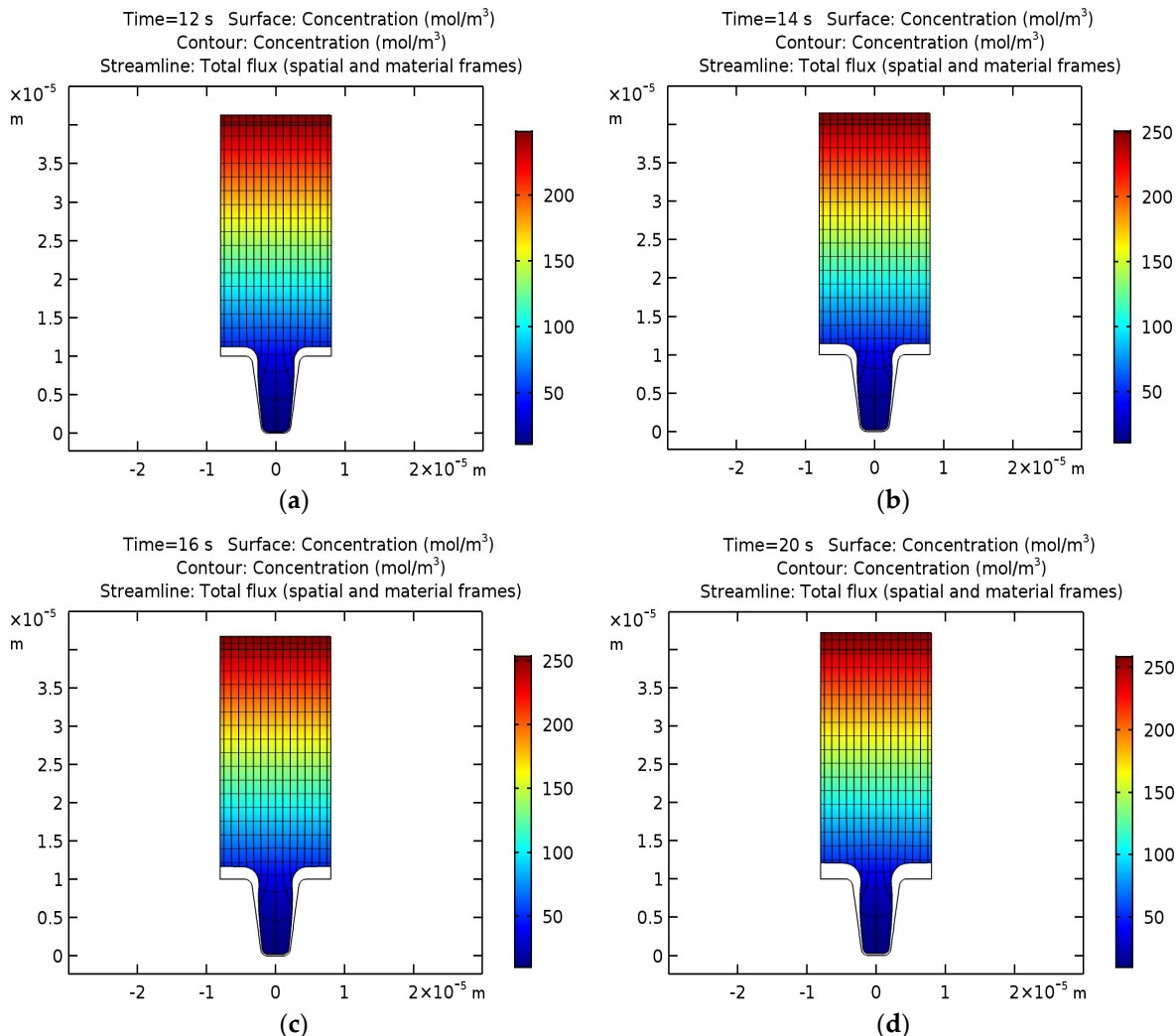

**Figure 3.** Concentrations of silver ions (mol/m$^3$), isopotential lines, current density lines, and electrode displacement in the cell after 12, 14, 16, and 20 s of operation.

It can be concluded that with an increase in the duration of the process, the cavity hole begins to narrow gradually due to the non-uniform thickness of the deposited layer, but differs significantly compared to the results for copper. The process of narrowing is slower, which may be associated with a lower initial concentration of silver. In this case, the optimal duration of the silver deposition process is 20 s.

Thus, the modeling of copper and silver deposition processes allows us to obtain preliminary data on the formation of the topology of contacts at the local metallization on the surface of printed circuit boards and other electronics products, without resorting to full-scale experiments.

## 4. Conclusion

The software COMSOL Multiphysics allowed us to perform the modeling of the processes of copper and silver electrodeposition in the trapezoidal profile groove.

The results obtained for the thicknesses of the deposited layers of copper and silver can optimize the deposition regimes. For example, it is advisable to carry out metal deposition in a periodic mode, changing the polarity of the electrode to the opposite. This should ensure the alignment of the concentration of metal ions in the depth of the groove, and thus reduce the thickness variation of the layers on the vertical walls.

**Author Contributions:** V.G.K. and L.V.K. provided the necessary theoretical materials, O.Y.E. performed experiments and analyzed data.

**Funding:** This research received no external funding.

**Conflicts of Interest:** The authors declare no conflict of interest.

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
