# Peer review of "Model of Nano-Metal Electroplating Process in Trapezoid Profile Groove"

_condensedmatter, doi:10.3390/condmat4010026_

Round 1

Reviewer 1 Report

Very good paper dealing with nanometal electroplating processes. The Authors show

very convincingly how modeling and simulation can effectively replace full-scale experiments

in obtaining data on the details of the metallization process.This method can be very useful also in the creation of hybrid systems Carbon-Metal at the nanostructured surface interface level.

Only one minor correction is needed in the introduction, i.e. 

"Modern technologies make it possible", instead of "Modern technnologies allow it possible"

Author Response

Pictures added that show the development of metal deposition at different operating times.

Corrected chapter titles and introduction.

so this sentence is not present, since the paragraph has been adjusted.

Reviewer 2 Report

The model strongly resemble a tutorial delivered with COMSOL multiphysics. The scientific problem is not presented in the manuscript. The methods are not described (it is not enough to report that you used COMSOL multiphysics). How do you used it? Which equations and approximations? Which number do you put in the model? There is a paper cited in the text which should describe the modelling strategy undertaken. There is still a complete lack of the definition of the modelling strategy. The presentation of the data is extremely short and there only a couple of plots which are difficult to read (they are just pasted from COMSOL interface).

Author Response

We have added an equation that describes the flux for each of the ions in the electrolyte.

Pictures added  that show the development of metal deposition at different operating times.

Reviewer 3 Report

The manuscript by O.Yu.Egorova et al. is devoted to computer simulation of the electrodeposition process on a cathode of a specific topography using COMSOL Multiphysics programme. The general idea of the paper is clear; it is connected with use of the computer simulation for prediction of the optimal electrodeposition time instead of the experimental trial.

However, I have to stress some imperfections in the presentation of the material.

- English language and style has to be improved. Some sentences are difficult for understanding (especially in the beginning of Introduction), some expressions are awkward. I would propose to use “aqueous electrolyte solution” instead of  “water electrolyte solution”, “flow of electric current” instead of “passage of electric current”, and similar. It is not clear what is meant by “release of a metal layer”, and how the structure defects move along the surface, forming a film (in the so called Main Part).

- It is not clear why the authors have chosen the particular shape of the part, which is subjected to the electroplating. Is this shape determined by the technological needs or by the standard options of the computer programme? It should be explained.

- It is not clear why the shape of the deposited layer is even in the case of copper deposition and uneven in the case of silver deposition. Why ion concentration in the solution and process duration affects not only thickness but also evenness of the layer in the case of silver?

- To my mind the authors have to perform or at least plan an experimental confirmation of the computer simulation results.

- The title Main Part is unacceptable.

- Is it implied that the paper parts Supplementary Materials, Acknowledgements and Author Contributions are not filled? Or they will be written later?

Author Response

In the library of the COMSOL Multiphysics model uses a rectangular groove. We wanted to compare how the process will flow when the geometry of the model and the material itself changes.

Experimental confirmation of results in the near future is not planned.

1.- It is not clear why the authors have chosen the particular shape of the part, which is subjected to the electroplating. Is this shape determined by the technological needs or by the standard options of the computer programme? It should be explained. (In the library of the COMSOL Multiphysics model uses a rectangular groove. We wanted to compare how the process will flow when the geometry of the model and the material itself changes.)

2.To my mind the authors have to perform or at least plan an experimental confirmation of the computer simulation results. (Experimental confirmation of results in the near future is not planned.)

3.- The title Main Part is unacceptable. (The title Main Part is replaced by The process of electrodeposition.)

4.- Is it implied that the paper parts Supplementary Materials, Acknowledgements and Author Contributions are not filled? Or they will be written later? (The paper parts Author Contributions are written.)

Round 2

Reviewer 3 Report

I agree that the manuscript can be published in the present form.